# Compliance with 24 h Movement Behavior Guidelines for Pregnant Women in Saudi Arabia: The Role of Trimester and Maternal Characteristics

**DOI:** 10.3390/healthcare12202042

**Published:** 2024-10-15

**Authors:** Saja Abdullah Alghamdi, Alawyah Alsalman, Om Kalthom Sowadi, Nada Khojah, Hadeel Saad, Bethany Barone Gibbs, Ghareeb Omar Alshuwaier, Abdullah Bandar Alansare

**Affiliations:** 1Department of Exercise Physiology, College of Sport Sciences and Physical Activity, King Saud University, King Khalid Rd, Riyadh 11451, Saudi Arabia; sajaalghamdi19@gmail.com (S.A.A.); omkalthom.sowadi@hotmail.com (O.K.S.); khojahnada@gmail.com (N.K.); hadeel.f.saad@gmail.com (H.S.); galshuwaier@ksu.edu.sa (G.O.A.); 2Department of Physical Education, College of Sport Sciences and Physical Activity, King Saud University, King Khalid Rd, Riyadh 11451, Saudi Arabia; alwyah2020@gmail.com; 3Department of Epidemiology and Biostatistics, School of Public Health, West Virginia University, Morgantown, WV 26506, USA; bethany.gibbs@hsc.wvu.edu

**Keywords:** pregnancy, physical activity, sleep, sedentary behavior, Arab, lifestyle

## Abstract

Background: Complying with 24 h movement behavior guidelines for pregnant women may prevent pregnancy complications. This single time point, cross-sectional investigation assessed compliance with the 24 h movement behavior guidelines in pregnant women in Saudi Arabia and examined the role of trimester and maternal characteristics. Methods: Pregnant women (n = 935; age = 30 ± 5.6 years; first trimester = 24.1%, second trimester = 33.9%, third trimester = 42.0%) self-reported their characteristics (nationality, region, degree, occupation, smoking status, health status, having children, previous birth). The short-version International Physical Activity Questionnaire, Sedentary Behavior Questionnaire, and Pittsburgh Sleep Quality Index Questionnaire measured moderate physical activity (MPA), sedentary behavior (SB), and sleep duration, respectively. Compliance with the 24 h movement behavior guidelines was reported using frequencies and percentages. Prevalence ratios compared the prevalence of compliance by trimester and maternal characteristics. Results: Approximately half of the participants did not comply with MPA or sleep duration guidelines (n = 524, [56.0%] and n = 424, [45.5%], respectively). In contrast, about two-thirds of participants (n = 648, [69.3%]) adhered to the SB guideline. Only 154 (16.5%) participants complied with all 3 24 h movement behavior guidelines. Pregnant women in their second trimester, living in Al-Ahsa Governorate, and currently smoking with a bachelor’s degree were the most likely to comply with the guidelines. Conclusions: These findings underscore the need for tailored efforts to promote healthy 24 h movement behavior guidelines for pregnant women in Saudi Arabia, especially early in pregnancy, while accounting for important maternal characteristics.

## 1. Introduction

Pregnancy is a physiological stress test for women [1]. Many physiological alternations, such as increased heart rate, plasma volume, total cortisol level, and urinary protein excretion, occur during pregnancy as normal adaptations to meet the demands of fetal development [2]. However, maladaptations to the physiological challenge of pregnancy can also occur in the form of complications, including gestational hypertension, gestational diabetes, or preeclampsia [1,3]. These complications may develop into long-term chronic health issues, contributing to increased morbidity and mortality among women [1,3]. A recommended strategic approach that may prevent these complications and its ramifications is the identification of modifiable risk factors associated with developing pregnancy complications and proactively altering them among pregnant women [4].

The 24 h movement behaviors, which refer to all movement behaviors (i.e., physical activity [PA], sedentary behavior [SB], and sleep) that occur throughout the day [5], are modifiable risk factors for the development and progression of pregnancy complications. For instance, low levels of PA, excessive SB, and/or poor sleep have been associated with gestational diabetes, gestational hypertension, and preeclampsia [6,7,8]. These unhealthy 24 h movement behaviors become more apparent with advanced pregnancy (i.e., three trimesters), and have been associated with characteristics of pregnant women. For instance, pregnant women tend to engage in less PA, accumulate higher SB, and have unfavorable sleep duration as their pregnancy progresses [9,10,11,12]. Furthermore, pregnant women with fewer children and who worked in sedentary occupations appear to accumulate higher SB while those who were white, employed, and married with higher education and had no history of pregnancy loss tended to engage in higher PA [13]. Hence, altering these behaviors to healthier ones may improve pregnancy outcomes and perhaps prevent any complications. Previous studies revealed that pregnant women who frequently perform PA accumulate a lesser amount of SB, and/or acquire an appropriate amount of sleep (8–9 h/day), have a lower risk of gestational diabetes mellitus, preeclampsia, and gestational hypertension, and lower postpartum depression [7,14,15]. As such, efforts have been initiated worldwide to promote healthy 24 h movement behaviors during and after pregnancy [16].

In 2020, the World Health Organization (WHO) released global movement behavior guidelines for pregnant and postpartum women [16], recommending that women should accumulate 150 min/week of moderate-intensity PA (MPA) and limit time spent in SB [16]. The Canadian 24 h movement guidelines, also released in 2020, added quantitative SB and sleep recommendations for adults, including pregnant women, who are recommended to accumulate <8 h/day of SB and sleep between 7 and 9 h/day [17,18]. These global initiatives emphasize the importance of healthy 24 h movement behaviors during pregnancy. However, the 24 h movement behavior guidelines for adults in Saudi Arabia lack specific recommendations for pregnant women mainly due to insufficient data on this population [19]. Furthermore, a common approach in most previous studies, locally and globally, is assessing only one or two components of these movement behaviors in pregnant women, limiting the holistic understanding of these modifiable risk factors in this population [11,20]. Given the high rates of pregnancy complications, such as gestational diabetes (24.2%) [21] and preeclampsia (5.37 per 10,000 pregnant women), in Saudi Arabia [22], exploring those who comply vs. do not comply with the 24 h movement behavior recommendations may contribute to constructing effective movement behavior interventions and guidelines tailored to pregnant women in Saudi Arabia.

Therefore, the main aims of this study were to (1) assess the compliance with the individual vs. combined 24 h movement behavior recommendations in pregnant women in Saudi Arabia and (2) examine the role of trimester and maternal characteristics in compliance with these guidelines among pregnant women. We hypothesized that most pregnant women in Saudi Arabia would not adhere to the MPA, SB, and/or sleep duration recommendations. We also hypothesized that compliance with the 24 h movement behavior guidelines would be lower in later vs. earlier trimesters and different across the maternal characteristics.

## 2. Materials and Methods

This study was reported following Strengthening the Reporting of Observational Studies in Epidemiology (STROBE) guidelines.

### 2.1. Study Design and Participants

This study employed a cross-sectional design with data collection at a single point in time for each participant. All data were collected between 3 July 2023 and 24 August 2023. To be enrolled in the study, participants had to meet the following criteria: (1) currently pregnant and (2) permanently living in Saudi Arabia. After providing informed consent, participants completed all questionnaires using a tablet computer, a technique that has been demonstrated to improve scorability and reduce missing data [23]. The study procedures were reviewed and approved by the institutional review board at King Saud University (No: KSU-HE-23-516).

In total, 952 participants were enrolled in the study. To mitigate the impact of outliers or potential problematic influential points, 17 participants were excluded due to missing height and/or weight (n = 1), unreasonable gestational age (e.g., 63 weeks) (n = 1), and unreasonable (i.e., >24 h/day) total SB (n = 2), sleep (n = 2), or PA (n = 11). The final analytic sample size was 935 pregnant women (first trimester = 24.1%, second trimester = 33.9%, third trimester = 42.0%).

### 2.2. Recruitment and Participants Selection

The recruitment and selection of the participants were performed using a thorough recruitment strategy as follows. First, all private and governmental hospitals and clinics offering obstetrics and gynecology services in six major cities in Saudi Arabia (Medina [West region], Riyadh [Central region], Al-Ahsa Governorate [East region], Jazan [South region], Jeddah [West region], and Taif [West region]) were reached. After obtaining the required permits, a research team member was stationed in waiting areas at the obstetrics and gynecology clinics. Therefore, all women visiting these clinics were directly reached, with the intention of including pregnant women with diverse demographic and socioeconomic statuses. All women who expressed interest in participation in the study were screened using the inclusion criteria. Thus, a wide spectrum of pregnant women who met the inclusion criteria were included in the study. This employed strategy has enhanced the diversity and repressiveness of pregnant women in Saudi Arabia.

### 2.3. Measurements

#### 2.3.1. Maternal Characteristics and Anthropometric Assessments

Standardized formal questions were used to obtain participants’ general characteristics and socioeconomic factors, including age, nationality (i.e., Saudi or non-Saudi), education (i.e., diploma or less, bachelor’s, master’s, or Ph.D.), occupation (i.e., student, unemployed, private sector, or governmental sector), current smoking status (i.e., yes or no), current health status (i.e., have chronic disease: yes or no), gestational age (i.e., weeks of pregnancy, used to determine pregnancy trimester), previous birth (i.e., yes or no), previous c-section birth (i.e., yes or no), and whether they had children (i.e., yes or no). The participants also self-reported their most recent measurements of body height (cm) and weight (kg).

##### PA

The Arabic version of the International Physical Activity Questionnaire short form (IPAQ-S) was used to measure participants’ PA [24]. The IPAQ-S is a multiple-question instrument (i.e., sex questions for different intensities of PA during the last 7 days), used to estimate time spent in light (LPA), moderate (MPA), vigorous (VPA), and moderate-to-vigorous PA (MVPA). The decision to use the IPAQ-S in this study was based on its ability to estimate time spent in different intensities of PA and its extensive global utilization [25,26,27]. In addition to its established reliability in pregnant women [28], using the IPAQ-S facilitates the comparison of this study’s findings with other international studies in pregnant women [28,29]. Additionally, time spent in MPA was used to evaluate whether pregnant women achieved the current PA recommendations (i.e., 150 min/week of MPA) [17,18,30].

##### Sleep

The Arabic version of the Pittsburgh Sleep Quality Index (PSQI) measures sleeping duration in pregnant women [31]. The PSQI includes several questions about sleep quality and/or duration, and its reliability in pregnant women was previously established [32,33]. In this study, only one question about sleep duration was used (i.e., “How many hours of actual sleep do you get at night?” in the past month). The answer to this question was used to assess if pregnant women achieved the current sleep duration guideline (i.e., 7–9 h/night) [18,30].

##### SB

The culturally adapted Arabic version of the Sedentary Behavior Questionnaire (SBQ) was implemented to measure time spent in SB [34]. The decision to use this multiple-item instrument to measure SB was to improve the precision of the SB estimate by capturing the time spent in different domains of SB [35]. Furthermore, the reliability of the SBQ in pregnant women was previously established [36]. The SBQ comprises 18 questions about time spent in different SBs on weekdays and weekends. Total SB was calculated using the following formula: total SB = [(total SB per weekday × 5) + (total SB per weekend day × 2)]/7 [37]. Total SB was used to classify pregnant women who adhered (i.e., accumulated < 8 h/day of total SB) vs. did not adhere (i.e., accumulated ≥ 8 h/day of total SB) to the total SB recommendation [18].

### 2.4. Statistical Analyses

To calculate the sample size representing females in Saudi Arabia, the Raosoft calculator, a web-based calculator, was used, which estimates the sample size needed for various study designs, including the cross-sectional design. Using the following information: the population size of 15,429,586 (according to the 2022 World Bank Data), confidence level of 95%, margin of error of 5%, and power of 80%, at least 385 women were needed. This study recruited more women to increase the precision and generalizability of the findings.

Maternal characteristics and time spent in each movement behavior for the included participants were reported as means with standard deviations, frequencies with percentages, or median with interquartile range, as appropriate. In addition, compliance with individual and combined movement behavior guidelines was reported using frequencies and percentages. Prevalence ratios (PRs) were estimated to compare compliance with the individual and combined movement behavior guidelines by maternal characteristics (i.e., nationality, region, degree, occupation, smoking status, health status, having children, and previous birth). Then, PRs were estimated to evaluate compliance with the individual and combined movement behavior guidelines by trimester with and without adjustment for significant maternal characteristics. Using the current existing recommendations, these estimations were completed with log-binomial regression models using Stata software (Stata 17.0 Version) [38]. The significance level was set at *p*-value < 0.05.

## 3. Results

Table 1 displays the characteristics of the included pregnant women in the current analyses. The average age of the participants was 30.0 ± 5.6 years. The majority of these pregnant women were Saudis (78.5%), with at least a bachelor’s degree (59.7%), non-smokers (97.8%), not currently working (76.8%), apparently healthy (91.2%), and likely to have at least one child (61.9%). Moreover, based on their reported gestational age, 24.1% were in their first trimester, 33.9% in their second trimester, and 42.0% in their third trimester. In addition, 36.7% of the participants revealed that this was their first pregnancy.

Table 2 and Figure 1 present compliance with the individual and combined 24 h movement behavior guidelines for the whole sample. Overall, the participants engaged in MPA for 2.1 ± 9.1 min/day, slept for 7.3 ± 2.1 h/day, and accumulated total SB for 6.8 ± 3.5 h/day. Nonetheless, only 44.0% of these pregnant women self-reported complying with the MPA recommendation (i.e., engaging in at least 150 min/week of MPA). Moreover, 54.5% of participants reported adhering to the sleep duration guideline (i.e., sleeping between 7 and 9 h/day). Noticeably, the majority of the pregnant women (69.3%) reported adhering to the total SB (i.e., <8 h/day) recommendations. Yet, compliance with at least two of these guidelines ranged between 23.5% and 38.0%, whereas only 16.5% of these pregnant women adhered to all three movement behavior guidelines.

When pregnant women were classified according to demographics (Table 3), only compliance with the MPA guideline was significantly (*p* < 0.05 for all) lower, particularly in those who lived in Jazan or Taif cities, had a diploma or less, and were current non-smokers. Furthermore, compliance with two of the 24 h movement behavior guidelines was significantly (*p* < 0.05 for all) lower among non-Saudi pregnant women who were a non-smoker, student, living in Riyadh, Jazan, or Taif cities with at least one previous child. Moreover, compliance with all 24 h movement behavior guidelines was significantly (*p* < 0.05 for all) lower among pregnant women who were living in Jazan, or Taif cities.

Table 4, Appendix A, and Figure 2 display compliance with the individual and combined 24 h movement behavior guidelines among pregnant women by trimesters. The adjusted PRs for significant demographics (i.e., nationality, region, degree, occupation, having a previous child, and smoking status) revealed that compliance with the MPA guideline was 37% higher in pregnant women in their second vs. first trimester (*p* < 0.05). PRs also found that compliance with the sleep duration recommendation was 4% higher in pregnant women in their third vs. first trimester (*p* < 0.05). However, compliance with the total SB recommendations was 21% lower in pregnant women in their third vs. first trimester (*p* < 0.05). Furthermore, PRs revealed that compliance with at least two or all 24 h movement behavior guidelines was higher in pregnant women in their second vs. first trimester (PR ranged from 40% to 54%; *p* < 0.05 for all). Lastly, simultaneous compliance with sleep duration and SB recommendations was 7% higher in pregnant women in their third vs. first trimester (*p* < 0.05).

## 4. Discussion

This unique cross-sectional investigation is the first to assess the prevalence of complying with the individual and combined 24 h movement behavior guidelines in pregnant women in Saudi Arabia. As hypothesized, the majority (56.0%) of these pregnant women did not adhere to the MPA guideline, whereas slightly under half (45.5%) of them did not meet the sleep duration guideline. In contrast to our hypothesis, about two-thirds (69.3%) of pregnant women complied with the SB recommendation. Only 16.5% of pregnant women adhered to the combined 24 h movement behavior guidelines. Maternal characteristics including nationality, region, education, occupation, having a previous child, or smoking status were associated with compliance with these movement behavior guidelines. When trimesters were considered, pregnant women in their second trimester were more likely to adhere to the individual and combined 24 h movement behavior guidelines than those in their first trimester. These findings underscore the need for tailored health promotion of 24 h movement behavior guidelines for pregnant women in Saudi Arabia, especially for those non-Saudi, in their first trimester, non-smokers, living in Riyadh, Jazan, or Taif cities, and with a diploma or less and having a previous child.

### 4.1. Strengths of the Study

This study has several aspects that strengthen our findings. Previous studies have largely evaluated one or two movement behaviors in pregnant women, limiting the comprehensive understanding of 24 h movement behaviors during pregnancy [11,20]. Herein, this study was the first to assess compliance with all 24 h movement behaviors (i.e., MPA, SB, and sleep) in pregnant women in their first, second, or third trimester in Saudi Arabia. As such, this study provides more comprehensive data on movement behaviors in pregnant women. Furthermore, in contrast to previous studies [11,20], this study included a large sample size from several cities across Saudi Arabia, which enhanced the generalizability of the findings.

### 4.2. Limitations of the Study

Still, a few limitations exist and should be considered when interpreting these findings. First, the majority of the included participants were relatively healthy pregnant women. As such, compliance with the 24 h movement behavior recommendations may be different in pregnant women with poorer health, especially those with high-risk pregnancies [39,40]. Moreover, although all 24 h movement behaviors were simultaneously evaluated, these measurements were completed using self-reported instruments, which are usually prone to recall and/or reporting errors [41]. For instance, pregnant women can unintentionally forget the exact number of minutes/hours they spend in PA or SB. Furthermore, self-report questionnaires can underestimate SB [41] and overestimate PA [42]. As such, the reported prevalence of compliance with SB and/or PA guidelines might have been under/overestimated. Thus, future research should consider using device-based approaches such as accelerometers to measure these movement behaviors in pregnant women. Another limitation is that the measurements were performed only at a single time-point during pregnancy while using a cross-sectional design, limiting our ability to infer longitudinal conclusions [43]. Hence, further longitudinal studies that assess compliance with the 24 h movement behavior guidelines in pregnant women throughout pregnancy are warranted to better understand the factors that influence patterns across pregnancy [44].

### 4.3. Compliance with Individual Movement Behavior Guidelines

Because participating in adequate prenatal PA is well-established to be associated with healthier pregnancy outcomes, compliance with the PA recommendation in pregnant women has been previously examined. For instance, a cohort study of pregnant women (n = 3868) in northern Sweden revealed that 52.9% of pregnant women did not adhere to the MPA recommendation (≥150 min/week) [45]. Another multi-ethnic cohort study of pregnant women (n = 555) in Norway demonstrated that 75% of pregnant women failed to meet the general MVPA recommendation (≥150 min/week) [46]. Interestingly, this prevalence was higher in the Middle Eastern (84%) and South Asian (86%) populations than in other ethnicities. Furthermore, a longitudinal study considered the trimester and noted that the prevalence of not complying with the general MVPA guideline was higher during the third trimester than during the second trimester (72% vs. 54%) in pregnant women (n = 46) in Iowa [47]. Herein, 56% of pregnant women in Saudi Arabia did not adhere to the MPA guideline. This pattern of insufficient PA was more apparent for pregnant women in their first (63.1%) and third (56%) trimester than for those in their second trimester (48%). Failure to adhere to the PA recommendation may pose significant health risks, such as gestational diabetes, hypertensive disorders of pregnancy, and excessive gestational weight gain, to pregnant women and their fetuses [1,3]. To improve pregnancy outcomes, the existing findings highlight the need for further efforts to specifically promote complying with the PA guideline by trimester and ethnicity, among pregnant women.

Although global and international SB guidelines urge pregnant women to limit time spent in SB [16,17,48], to perhaps ≤ 8 h/day [18], several reports indicate low compliance with these recommendations. For example, a recent Canadian survey observed that pregnant women (n = 1625) spent an average of 9.5 h/day in SB [49]. Similarly, Spanish pregnant women (n = 134) were reported to spend an average of 8.5 h/day in SB [50]. In one American study using device-based measurement (n = 120), small SB differences were detected between the first, second, and third trimesters (65.0% time/day, 63.0% time/day, and 63.3% time/day, respectively), although all trimesters averaged approximately 9 h per day [10]. Adding to these findings, this study showed that 30.7% of pregnant women in Saudi Arabia accumulated >8 h/day of SB. This prevalence was higher among pregnant women in their first and third trimester (31.6% and 30.8%) than those in their second trimester (30.0%). Although adherence to the SB guideline is better than that to the PA guideline, continuous efforts are necessary to improve pregnancy outcomes.

Acquiring an adequate sleep duration is a crucial factor for optimal pregnancy outcomes. However, worldwide data suggest low adherence to the sleep duration recommendation during pregnancy. For instance, a cross-sectional investigation of Chinese pregnant women (n = 2345) revealed that 44.8% had an inadequate sleep duration [51]. Moreover, recent national data on American pregnant women (n = 2349) indicated that 29% of the participants did not achieve the sleep duration recommendation [52]. Consistently, a longitudinal study observed sleep duration differences across trimesters where American pregnant women tended to accumulate less sleeping duration during their third trimester (mean = 6.5 h/day) than during their first and second trimester (mean = 7.1 h/day for both) [10]. Aligning with these findings, the current study showed that 45.5% of pregnant women in Saudi Arabia did not adhere to the sleep duration guideline. In addition, significant differences were detected between trimesters, such that the prevalence of not complying with the sleep duration recommendation was higher in pregnant women in their first trimester than those in their second or third trimester (52.4% vs. 44.5% and 42.2%, respectively). Taken together, the existing evidence suggests that compliance with the sleep duration guideline remains low and may vary by ethnicity or country, indicating the need for further international collaborations to properly promote the sleep duration recommendation for pregnant women.

### 4.4. Compliance with the Combined 24 h Movement Behavior Guidelines

The 24 h movement behavior paradigm is increasingly adopted in health research and guidelines for all groups due to its holistic approach and significant associations with health outcomes [53]. However, this area remains largely unexplored within the context of pregnancy. To the best of our knowledge, this study is the first investigation that contributes to understanding the angle of compliance with the 24 h movement behaviors in pregnant women, especially among women in Saudi Arabia. Unfortunately, 83.5% of pregnant women in Saudi Arabia did not adhere to the combined 24 h movement behavior recommendations. In addition, this prevalence was higher for pregnant women in their first (87.1%) and third (85.5%) trimester vs. second trimester (78.5%). Only two previous studies assessed the prevalence of adherence to the combined 24 h movement behaviors in female adults. First, the Thai national time-use survey, which included 23,755 women, reported that 76.7% of all women (including pregnant women) did not meet these recommendations [54]. Moreover, a multi-national cross-sectional study of Latin American women (n = 1090) revealed that 98.8% of women (including pregnant women) failed to adhere to the 24 h movement behavior guidelines [55]. Further research is needed to understand how 24 h movement behaviors combine to affect pregnancy health and how to promote achievement of the holistic recommendations for pregnant women.

### 4.5. Maternal Characteristics

Our study also revealed that compliance with the 24 h movement behavior recommendations among pregnant women in Saudi Arabia may be influenced by maternal characteristics including nationality, region, education, occupation, having a previous child, or smoking status. These findings are in alignment with previous studies among pregnant women in different countries. For instance, it was found that having a previous child and education level affected compliance with the MVPA guidelines among Spanish pregnant women [50]. Likewise, the number of previous children, education level, occupational status or type, and ethnicity impacted the time spent in SB and PA among American pregnant women [13]. Moreover, occupational status and type and ethnicity were observed to influence the sleep duration among New Zealand pregnant women [56]. Together, the currently existing data suggest that maternal characteristics have an important role in compliance with the 24 h movement behavior guidelines among pregnant women. As such, maternal characteristics should be considered in future recommendations and behavioral interventions that target 24 h movement behaviors in pregnant women to achieve optimal outcomes.

### 4.6. Clinical Significance

Compliance with the 24 h movement behavior guidelines for pregnant women may prevent several persistent maladaptations associated with pregnancy [7,14,15]. Accordingly, global efforts promoting these recommendations have been raised [16,17,18]. By focusing on compliance with the 24 h movement behavior guidelines for pregnant women, this study offers unique findings about pregnant women in Saudi Arabia. The majority of pregnant women in Saudi Arabia tend not to adhere to individual or combined 24 h movement behavior recommendations. Maternal characteristics played a significant role in compliance with these guidelines. Furthermore, this pattern was more apparent in women in their first or third trimester vs. second trimester. These results highlight the current global and international efforts urging pregnant women to comply with the 24 h movement behavior guidelines. Thus, further efforts are warranted to promote the 24 h movement behavior recommendations for pregnant women in Saudi Arabia, with specific strategies for each pregnancy stage. Importantly, smoking status, region, and education appeared to be the most influential maternal characteristics on compliance with 24 h movement behaviors. Therefore, planning interventions that aim to promote healthy 24 h movement behaviors in pregnant women should consider these factors.

## 5. Conclusions

To summarize, this study was the first study in Saudi Arabia that examined compliance with the individual vs. combined 24 h movement behavior guidelines in pregnant women in Saudi Arabia and evaluated the role of trimesters and maternal characteristics. The findings revealed that most pregnant women tended not to adhere to the 24 h movement behavior recommendations, especially pregnant women in their first trimester. Maternal characteristics including nationality, region, education, occupation, having a previous child, and smoking status have a significant role in compliance with these guidelines. These results are of significant importance as global and local efforts aim to increase adherence to healthy movement behaviors during pregnancy to prevent complications. Although future longitudinal research is essential, the current results emphasize the need for tailored health promotion of 24 h movement behavior guidelines for pregnant women in Saudi Arabia, especially for those in their first trimester, non-smokers, those living in the cities of Jazan or Taif, and with a diploma or less.

## Figures and Tables

**Figure 1 healthcare-12-02042-f001:**
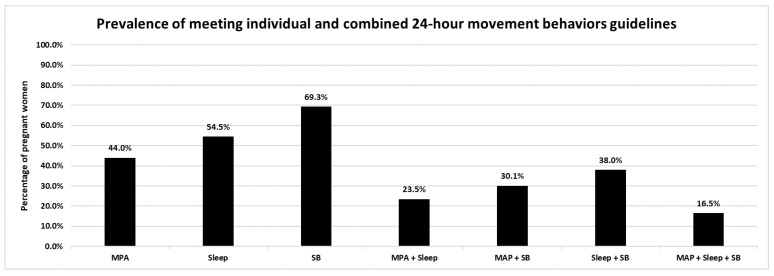
Compliance with the 24 h movement behaviors guidelines. MPA; moderate physical activity, SB; sedentary behavior.

**Figure 2 healthcare-12-02042-f002:**
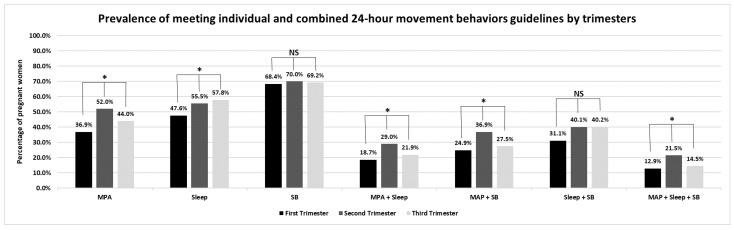
Compliance with the 24 h movement behaviors guidelines by trimester. * indicates a significant prevalence ratio. NS indicates a non-significant prevalence ratio. MPA; moderate physical activity, SB; sedentary behavior.

**Table 1 healthcare-12-02042-t001:** Characteristics of the included participants (n = 935).

Characteristic	Mean ± SD, n (%), or Median (IQR)
Age (years old)	30.0 ± 5.6
Height (cm)	158.7 ± 6.4
Weight (kg)	69.1 ± 14.7
Nationality	
Saudi	734 (78.5%)
Non-Saudi	200 (21.4%)
Did not Report	1 (0.1%)
Region of Saudi Arabia	
Medina (West)	249 (26.6%)
Riyadh (Center)	172 (18.4%)
Al-Ahsa Governorate (East)	214 (22.9%)
Jazan (South)	102 (10.9%)
Jeddah (West)	102 (10.9%)
Taif (South West)	96 (10.3%)
Degree	
Diploma or Less	386 (41.3%)
Bachelor	520 (55.6%)
Master’s	21 (2.2%)
PhD	8 (0.9%)
Occupation	
Governmental Sector	68 (7.3%)
Private Sector	94 (9.9%)
Currently Unemployed	718 (76.8%)
Student	56 (6.0%)
Currently Smoking	
No	914 (97.8%)
Yes	21 (2.2%)
Chronic Disease	
No	853 (91.2%)
Yes	82 (8.8%)
Have Children	
No	356 (38.1%)
Yes	579 (61.9%)
Gestational Age	
First Trimester	225 (24.1%)
Second Trimester	317 (33.9%)
Third Trimester	393 (42.0%)
Previous Birth	
No	343 (36.7%)
Yes	592 (63.3%)

cm; centimeter, IQR; interquartile range, kg; kilogram, n; number, SD; standard deviation.

**Table 2 healthcare-12-02042-t002:** Compliance with the individual and combined 24 h movement behaviors guidelines among pregnant women.

Variable	MPA (min/week)	Sleep Duration (h/day)	Total SB (h/day)	Combined 24 h Movement Behaviors			
Mean ± SD	210.2 ± 325.9	7.3 ± 2.1	6.8 ± 3.5	N/A			
**Adhered to**	**MPA** **Guideline** **(150 min/week)**	**Sleep Duration** **Guideline** **(7–9 h/day)**	**Total SB** **Guideline** **(<8 h/day)**	**MPA + Sleep Duration** **Guidelines**	**MPA + Total SB** **Guidelines**	**Sleep Duration + Total SB** **Guidelines**	**MPA+ Sleep Duration + Total SB** **Guidelines**
	**n** **(%)**	**n** **(%)**	**n** **(%)**	**n** **(%)**	**n** **(%)**	**n** **(%)**	**n** **(%)**
Yes	411 (44.0%)	510 (54.5%)	648 (69.3%)	220 (23.5%)	281 (30.1%)	355 (38.0%)	154 (16.5%)
No	524 (56.0%)	425 (45.5%)	287 (30.7%)	715 (76.5%)	654 (69.9%)	580 (62.0%)	781 (83.5%)

h; hour, min; minute, MPA; moderate physical activity, n; number, SB; sedentary behavior, N/A; not applicable, SD; standard deviation.

**Table 3 healthcare-12-02042-t003:** Compliance with the individual 24 h movement behaviors guidelines by maternal characteristics.

Characteristic	Complying MPA Guideline (150 min/week)	Complying Sleep Duration Guideline (7–9 h/day)	Complying Total SB Guideline (<8 h/day)	Adhered to MPA + Sleep Duration Guidelines	Adhered to MPA + Total SB Guidelines	Adhered to Sleep Duration + Total SB Guidelines	Adhered to MPA + Sleep Duration + Total SB Guidelines
	PR (95% CI)	PR (95% CI)	PR (95% CI)	PR (95% CI)	PR (95% CI)	PR (95% CI)	PR (95% CI)
	Nationality						
Non-Saudi	Reference	Reference	Reference	Reference	Reference	Reference	Reference
Saudi	1.20 (0.99–1.46)	0.91 (0.80–1.04)	0.99 (0.90–1.10)	1.00 (0.76–1.33)	**1.32** **(1.01–1.73)**	0.90 (0.75–1.09)	1.13 (0.78–1.63)
	Region						
Medina	Reference	Reference	Reference	Reference	Reference	Reference	Reference
Riyadh	0.92 (0.85–1.00)	1.00 (0.95–1.06)	1.02 (0.98–1.07)	**0.54** **(0.36–0.81)**	0.77 (0.54–1.08)	1.13 (0.89–1.44)	0.66 (0.41–1.07)
Al-Ahsa Governorate	**1.47** **(1.23–1.74)**	0.90 (0.76–1.07)	1.09 (0.97–1.23)	1.13 (0.86–1.49)	**1.68** **(1.32–2.14)**	1.02 (0.81–1.30)	1.34 (0.94–1.90)
Jazan	**0.61** **(0.43–0.87)**	0.89 (0.71–1.11)	0.98 (0.83–1.16)	**0.34** **(0.18–0.63)**	**0.52** **(0.31–0.96)**	0.85 (0.61–1.18)	**0.32** **(0.14–0.72)**
Jeddah	1.22 (0.97–1.54)	1.08 (0.89–1.31)	1.09 (0.94–1.27)	0.95 (0.66–1.38)	**1.39** **(1.02–1.91)**	1.11 (0.84–1.48)	1.01 (0.62–1.63)
Taif	**0.67** **(0.48–0.95)**	0.94 (0.76–1.17)	1.05 (0.90–1.23)	**0.47** **(0.27–0.80)**	0.67 (0.42–1.06)	1.01 (0.75–1.38)	**0.51** **(0.26–0.99)**
	Degree						
Diploma or Less	Reference	Reference	Reference	Reference	Reference	Reference	Reference
Bachelor	**1.20** **(1.03–1.40)**	1.02 (0.90–1.15)	0.99 (.92–1.09)	1.18 (0.93–1.50)	1.16 (0.95–1.43)	1.10 (.93–1.31)	1.21 (0.90–1.64)
Master’s	1.45 (0.98–2.14)	0.56 (0.27–1.04)	0.89 (0.63–1.26)	0.94 (0.60–1.48)	1.18 (0.89–1.57)	0.73 (.47–1.14)	0.98 (0.57–1.68)
PhD	0.63 (0.19–2.12)	1.38 (0.91–2.08)	1.08 (0.72–1.62)	.58 (0.09–3.67)	0.46 (0.07–2.87)	1.74 (0.99–3.02)	0.85 (0.13–5.38)
	Occupation						
Student	Reference	Reference	Reference	Reference	Reference	Reference	Reference
Private Sector	0.80 (0.55–1.17)	1.33 (0.94–1.89)	1.02 (0.79–1.31)	1.26 (0.67–2.38)	1.04 (0.61–1.79)	**1.68** **(1.00–2.80)**	1.91 (0.81–4.49)
Currently Unemployed	0.92 (0.70–1.23)	1.31 (0.96–1.78)	1.13 (0.92–1.39)	1.23 (0.72–2.13)	1.15 (0.74–1.80)	1.56 (0.98–2.48)	1.56 (0.72–3.38)
Governmental Sector	0.92 (0.76–1.12)	1.01 (0.83–1.24)	1.04 (0.91–1.19)	0.90 (0.43–1.88)	0.99 (0.55–1.78)	1.29 (0.73–2.29)	1.24 (0.47–3.26)
	Smoking Status						
Non-Current Smokers	Reference	Reference	Reference	Reference	Reference	Reference	Reference
Current Smokers	**1.42** **(1.01–2.00)**	1.05 (0.72–1.53)	0.96 (.71–1.30)	1.00 (0.76–1.33)	**1.32** **(1.01–1.73)**	0.90 (0.75–1.09)	1.13 (0.78–1.63)
	Health Status						
Do Not Have a Chronic Disease	Reference	Reference	Reference	Reference	Reference	Reference	Reference
Have a Chronic Disease	0.85 (0.64–1.13)	1.01 (0.82–1.24)	1.12 (0.99–1.27)	0.93 (0.60–1.42)	1.06 (0.76–1.48)	1.10 (0.84–1.45)	1.12 (0.69–1.82)
	Children						
Do Not Have a Child	Reference	Reference	Reference	Reference	Reference	Reference	Reference
Have a Child	0.91 (0.78–1.05)	1.07 (0.95–1.21)	0.98 (0.90–1.07)	0.87 (0.69–1.10)	**0.81** **(0.67–0.99)**	1.01 (0.85–1.20)	0.80 (0.60–1.07)
	Previous Birth						
Do Not Have a Previous Birth	Reference	Reference	Reference	Reference	Reference	Reference	Reference
Have at Least One Previous Birth	0.88 (0.76–1.02)	1.04 (0.92–1.18)	0.99 (0.91–1.08)	0.89 (0.70–1.12)	0.83 (0.68–1.01)	1.00 (0.84–1.19)	0.86 (0.64–1.15)

Bold indicates significant prevalence ratio. CI; confident interval, h; hour, min; minute, MPA; moderate physical activity, PR; prevalence ratio, SB; sedentary behavior.

**Table 4 healthcare-12-02042-t004:** Compliance with the individual and combined 24 h movement behaviors guidelines by trimester with adjustment for demographics.

Trimesters	Adhered to MPA Guideline (150 min/week)	Adhered to Sleep Duration Guideline (7–9 h/day)	Adhered to Total SB Guideline (<8 h/day)	Adhered to MPA + Sleep Duration Guidelines	Adhered to MPA + Total SB Guidelines	Adhered to Sleep Duration + Total SB Guidelines	Adhered to MPA + Sleep Duration + Total SB Guidelines
	PR (95% CI)	PR 95%(CI)	PR 95%(CI)	PR 95%(CI)	PR 95%(CI)	PR 95%(CI)	PR 95%(CI)
First (n = 225)	Reference	Reference	Reference	Reference	Reference	Reference	Reference
Second (n = 317)	**1.37** **(1.13–1.66)**	1.05 (0.92–1.19)	0.97 (0.87–1.09)	**1.43** **(1.04–1.98)**	**1.40** **(1.08–1.83)**	1.08 (0.91–1.29)	**1.54** **(1.08–2.30)**
Third (n = 393)	1.14 (0.92–1.41)	**1.04** **(1.02–1.06)**	**0.79** **(0.74–0.84)**	1.20 (0.86–1.69)	1.12 (0.85–1.49)	**1.07** **(1.03–1.10)**	1.21 (0.79–1.84)

Bold indicates significant prevalence ratio. CI; confident interval, h; hour, min; minute, MPA; moderate physical activity, PR; prevalence ratio, SB; sedentary behavior. All models were adjusting for nationality, region, degree, occupation, having a previous child, and smoking status.

## Data Availability

The data are available upon requesting from the corresponding author.

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
