# Peer review of "Compliance with 24 h Movement Behavior Guidelines for Pregnant Women in Saudi Arabia: The Role of Trimester and Maternal Characteristics"

_healthcare, 2024, doi:10.3390/healthcare12202042_

Round 1
Reviewer 1 Report
Comments and Suggestions for Authors
I really enjoyed reading the manuscript titled: ‘Compliance with 24-hour Movement Behavior Guidelines for Pregnant Women in Saudi Arabia: The Role of Trimester and Maternal Characteristics’. The topic of the manuscript is very important among public health problems connected with the health of pregnant women and their offspring. Such diagnosis are particulary important to the development of preventive programmes aimed at increasing the awareness of pregnant and the promotion of prohealth changes in their lifestyle.
The research approach, the interpretations, conclusions and discussion are rather comprehensive. There are some concerns, which the authors should address to:
1. Introduction : The expression ’24-hour movement behavior’ should be explained clearly . Also the reference to the sources of this nomenclature should be added.
2. Materials and Method:
a. Study desing and participans:
- You write about the inclusion criteria. What about health condition of participants: have you decided to examine all interested pregnant women or were there any additional inclusion/exclusion criteria? I am wondering in particular about high-risk pregnancy women who can not be engaded in physical activity and have specific movement limitations which makes them unable to meet the WHO PA and SB guidelines. How have you solve this problem? Additionaly IPAQ procedure assumes the examination of typical week – in the case of your study a typical week of examined pregnant women. Have you fulfilled this criterion?
- The information about the numbers of study group is missing. It is only in the Abstract. Please complete it.
- The process of the final study group formation should be described (drop-outs connected with inclusion/exclusion criteria or a flow-chart).
b. Measurments:
- IPAQ description: Short version gives the possibility to get information about both PA (MPA, VPA) and SB. Why have you decided to use additional questionaire (SBQ).
- the information about the general reliability of used scales (IPAQ, PSQI, SBQ) as well as in the study group are missing.
- please complete the information about the form of the scales used in the study (paper or electronic).
3. Statistical analysis: the information about the normality of distribution of the examined variables is missing.
4. Results:
- information in the first paragraph should be added to the previous section. The process of the formation of the final study group should be described in the Study design and participants’ section.
- Fig. 2: Why have you decided to compare the prevalence of compliance with the single and all guidelines between trimesters instead of making the comparison between the nominal variables (MPA weekly volume, SB weekly volume, sleep duration) using parametric or non-parametric intergroup comparison tests? (variance analisys).
Author Response
Reviewer 1:
I really enjoyed reading the manuscript titled: ‘Compliance with 24-hour Movement Behavior Guidelines for Pregnant Women in Saudi Arabia: The Role of Trimester and Maternal Characteristics’. The topic of the manuscript is very important among public health problems connected with the health of pregnant women and their offspring. Such diagnoses are particularly important to the development of preventive programs aimed at increasing the awareness of pregnant and the promotion of health changes in their lifestyle.
The research approach, interpretations, conclusions, and discussion are rather comprehensive. There are some concerns, which the authors should address to:
Response: We would like to thank the reviewer for their valuable comments and time spent reviewing our manuscript. Each comment was carefully considered. Please see the responses below.
- Comment: Introduction : The expression ’24-hour movement behavior’ should be explained clearly . Also the reference to the sources of this nomenclature should be added.
Response:
Thank you for your important comment. As the reviewer has suggested, the expression ’24-hour movement behavior’ was explained clearly and the relevant reference was added. We believe that these changes have significantly strengthened the clarity and comprehensiveness of the introduction. Please see the modified sentences in the introduction section.
"The 24-hour movement behaviors, which refer to all movement behaviors (i.e., physical activity [PA], sedentary behavior [SB], and sleep) that occur throughout the day (5), are modifiable risk factors for the development and progression of pregnancy complications. For instance, low levels of PA, excessive SB, and/or poor sleep have been associated with gestational diabetes, gestational hypertension, and preeclampsia (6-8)."
- Comment: Materials and Method: a. Study desing and participans: You write about the inclusion criteria. What about health condition of participants: have you decided to examine all interested pregnant women or were there any additional inclusion/exclusion criteria? I am wondering in particular about high-risk pregnancy women who can not be engaded in physical activity and have specific movement limitations which makes them unable to meet the WHO PA and SB guidelines. How have you solve this problem? Additionaly IPAQ procedure assumes the examination of typical week – in the case of your study a typical week of examined pregnant women. Have you fulfilled this criterion?
Response:
Thank you for your crucial and critical point. We agree with the reviewer that several important health conditions of participants, such as high-risk pregnancy, can potentially affect their adherence to PA and SB guidelines. To address this, we collected information about the participants' health status, including any existing chronic diseases. Only 8.8% of participants reported having chronic diseases, and high-risk pregnancy was not identified as a factor among our sample. Despite this, we have acknowledged this as a significant limitation in our limitations section.
"As such, compliance with the 24-hour movement behavior recommendations may be different in pregnant women with poorer health, especially those with high-risk pregnancies (39, 40)."
Regarding the IPAQ procedure, which assumes the examination of a typical week, we explicitly instructed participants to answer each question while considering their typical week's activities. Therefore, we believe that this criterion was adequately fulfilled.
- Comment: Materials and Method: The information about the numbers of study group is missing. It is only in the Abstract. Please complete it.
Response:
Thank you for your comment. As the reviewer has suggested, the numbers of the study's groups were added to the Materials and Methods section.
"In total, 952 participants were enrolled in the study. To mitigate the impact of outliers or potential problematic influential points, 17 participants were excluded due to missing height and/or weight (n=1), unreasonable gestational age (e.g., 63 weeks) (n=1), and un-reasonable (i.e., >24 hours/day) total SB (n=2), sleep (n=2), or PA (n=11). The final analytic sample size was 935 pregnant women (first trimester=24.1%, second trimester=33.9%, third trimester=42.0%)."
- Comment: Materials and Method: The process of the final study group formation should be described (drop-outs connected with inclusion/exclusion criteria or a flow-chart).
Response:
Thank you for your comment. As the reviewer has suggested, the process of the final study group formation was described in the Materials and Methods section.
"In total, 952 participants were enrolled in the study. To mitigate the impact of outliers or potential problematic influential points, 17 participants were excluded due to missing height and/or weight (n=1), unreasonable gestational age (e.g., 63 weeks) (n=1), and un-reasonable (i.e., >24 hours/day) total SB (n=2), sleep (n=2), or PA (n=11). The final analytic sample size was 935 pregnant women (first trimester=24.1%, second trimester=33.9%, third trimester=42.0%)."
- Comment: Materials and Method: b. Measurments: IPAQ description: Short version gives the possibility to get information about both PA (MPA, VPA) and SB. Why have you decided to use additional questionaire (SBQ).
Response:
Thank you for raising this crucial point. We acknowledge that while the IPAQ includes a single question about SB, it may not comprehensively capture all domains of SB, such as leisure and occupational activities. To address this limitation, we supplemented the IPAQ with a more detailed multiple-item SB questionnaire (SBQ). This approach allowed us to obtain a more precise estimate of total SB by considering various domains. We have provided further clarification on this point in the materials and methods section.
"The decision to use this multiple-item instrument to measure SB was to improve the precision of the SB estimate by capturing time spent in different domains of SB (35). Furthermore, the reliability of the SBQ in pregnant women was previously established (36)."
- Comment: Materials and Method: the information about the general reliability of used scales (IPAQ, PSQI, SBQ) as well as in the study group are missing.
Response:
Thank you for your comment. As the reviewer has suggested, information about the reliability of the used scales in pregnant women was added.
"In addition to its established reliability in pregnant women (28), using the IPAQ-S facilitates the comparison of this study's findings with other international studies in pregnant women (28, 29)."
"The PSQI includes several questions about sleep quality and/or duration, and its reliability in pregnant women was previously established (32, 33)."
"Furthermore, the reliability of the SBQ in pregnant women was previously established (36)."
- Comment: Materials and Method: please complete the information about the form of the scales used in the study (paper or electronic).
Response:
Thank you for this comment. As the reviewer has suggested, information about the form of the scales used in the study was added.
"After providing informed consent, participants completed all questionnaires using a tablet computer, a technique that has been demonstrated to improve scorability and reduce missing data (23)."
- Comment: Statistical analysis: the information about the normality of distribution of the examined variables is missing.
Response:
Thank you for your critical point. The statistical analyses used in this study employed log-binomial regression models to calculate prevalence ratios. It is important to note that log-binomial regression models do not require the assumption of normality for the outcome variable, as the outcome in this case is binary. Moreover, while the normality assumption can be violated with smaller sample sizes, our relatively large sample size of 935 participants suggests that our variables were likely normally distributed.
- Comment: Results: information in the first paragraph should be added to the previous section. The process of the formation of the final study group should be described in the Study design and participants’ section.
Response:
Thank you for your comment. As the reviewer has suggested, the process of the formation of the final study group was described in the study design and participants’ section.
- Comment: Results: Fig. 2: Why have you decided to compare the prevalence of compliance with the single and all guidelines between trimesters instead of making the comparison between the nominal variables (MPA weekly volume, SB weekly volume, sleep duration) using parametric or non-parametric intergroup comparison tests? (variance analysis).
Response:
Thank you so much for your insightful question regarding our choice of analysis. This study prioritizes assessing compliance with 24-hour movement behaviors guidelines in pregnant women, identifying groups that meet or do not meet the recommendations. This informs targeted interventions. Analyzing compliance rates by trimester helps to understand which populations might need additional support. While examining MPA, SB, and sleep duration could offer insights, the focus of the current manuscript is on compliance patterns to guide effective interventions.
Reviewer 2 Report
Comments and Suggestions for Authors
While this manuscript addresses an important topic, a potentially important concern of movement behaviors amongst pre-conception and pregnant Saudi women, is also majorly flawed in its design and execution, causing major concerns regarding the validity as well as generalizability of outcomes. Taken together, these deficiencies greatly reduce the value of the study to the scientific community and raise doubts as to its suitability for publication in its present form.
Major Concerns:
1. Insufficient Novelty and Research Gap:
The manuscript fails to clearly articulate the novelty of the research. While compliance with movement behavior guidelines is an important topic, the authors do not sufficiently demonstrate how this study adds value to the existing body of literature, either globally or within the specific Saudi Arabian context.
Without a clear statement of the research gap or a comprehensive comparison with prior studies, the manuscript lacks the novelty.
2. Methods
The study's cross-sectional design limits its ability to provide insights into the cause-and-effect relationships between movement behaviors and maternal outcomes. Furthermore, the authors acknowledge the limitations of using self-reported data, yet they do not adequately discuss how this introduces bias into the results or suggest ways to mitigate this in future research.
The recruitment process is poorly described, raising concerns about potential selection bias. There is no clear explanation of how the sample was representative of pregnant women in Saudi Arabia, nor how it addresses potential confounding factors such as socio-economic disparities.
The study relies solely on self-reported questionnaires, which are known to be prone to recall and reporting biases. In the absence of objective measures (e.g., accelerometers), the validity of the findings is questionable.
3. Lack of Depth in Statistical Analysis:
While the authors use prevalence ratios and regression models, there is limited explanation of how the models were adjusted for potential confounders. Moreover, with only 21 current smokers in the sample, it is highly questionable whether meaningful statistical conclusions can be drawn regarding smoking status.
The paper would benefit from more advanced statistical techniques, such as multivariate analyses or sensitivity tests, to strengthen the reliability of the findings. The current analysis does not seem rigorous enough to support the conclusions drawn.
3. Weak Discussion and Implications:
The discussion is not well developed and does not sufficiently connect the evidence to implications for practice or policy. While the authors acknowledge that further interventions should be considered, they offer no strategies to integrate them into existing rehabilitation practices in Saudi Arabia. Moreover, they do not sufficiently relate their results to existing studies (especially other studies in the same facility or on similar populations). This diminishes the case for generalizability of their results.
4. Conclusion Lacks Impact:
The conclusion is limited to interpretation of results already presented in earlier sections, without new insights or concrete suggestions of where the results matter for public health practice. The study adds virtually nothing to the armamentarium of policy or clinical recommendations.
Comments on the Quality of English LanguageOverall, the English used in the manuscript is acceptable but needs more clarity and flow, as well as professionalism. Although the authors are usually good at expressing their ideas, there are a few things that, in my opinion, make readability hard.
Author Response
Reviewer 2:
While this manuscript addresses an important topic, a potentially important concern of movement behaviors amongst pre-conception and pregnant Saudi women, is also majorly flawed in its design and execution, causing major concerns regarding the validity as well as generalizability of outcomes. Taken together, these deficiencies greatly reduce the value of the study to the scientific community and raise doubts as to its suitability for publication in its present form. Major Concerns:
Response: We would like to thank the reviewer for their valuable comments and time spent reviewing our manuscript. Each comment was carefully considered. Please see the responses below.
- Comment: Insufficient Novelty and Research Gap: The manuscript fails to clearly articulate the novelty of the research. While compliance with movement behavior guidelines is an important topic, the authors do not sufficiently demonstrate how this study adds value to the existing body of literature, either globally or within the specific Saudi Arabian context. Without a clear statement of the research gap or a comprehensive comparison with prior studies, the manuscript lacks the novelty.
Response:
Thank you for your comment. The novelty and significance of our study lie in two key aspects. First, existing 24-hour movement behavior guidelines in Saudi Arabia lack specific recommendations for pregnant women due to insufficient data on their movement behaviors. Given the global emphasis on improving women's health, including pregnant women, our research fills a crucial knowledge gap by providing data on the prevalence of compliance with 24-hour movement behaviors in pregnant women in Saudi Arabia.
Second, most previous studies, both locally and globally, have assessed only one or two components of 24-hour movement behaviors in pregnant women. Our study's comprehensive evaluation of all three components (physical activity, sedentary behavior, and sleep) offers a more holistic understanding of the 24-hour movement behaviors, their patterns, and associated factors.
As the reviewer implied, we incorporated further clarifications regarding the study's novelty and significance into the introduction. Please see the revised introduction.
"However, the 24-hour movement behavior guidelines for adults in Saudi Arabia lack specific recommendations for pregnant women mainly due to insufficient data on this population (19). Furthermore, a common approach in most previous studies, locally and glob-ally, is assessing only one or two components of these movement behaviors in pregnant women, limiting the holistic understanding of these modifiable risk factors in this population (11, 20)."
- Comment: Methods: The study's cross-sectional design limits its ability to provide insights into the cause-and-effect relationships between movement behaviors and maternal outcomes. Furthermore, the authors acknowledge the limitations of using self-reported data, yet they do not adequately discuss how this introduces bias into the results or suggest ways to mitigate this in future research.
Response:
Thank you for your valuable and critical point. We acknowledge the limitations of cross-sectional design and self-report instruments, including their potential to overestimate physical activity and underestimate sedentary behavior. Additionally, self-report instruments may be subject to random, systematic, or reporting errors. As the reviewer has suggested, we have added a more detailed discussion of these biases to the limitations section. We have also outlined potential strategies for future studies to address and mitigate these limitations.
"As such, compliance with the 24-hour movement behavior recommendations may be different in pregnant women with poorer health, especially those with high-risk pregnancies (39, 40). Moreover, although all 24-hour movement behaviors were simultaneously evaluated, these measurements were completed using self-reported instruments, which are usually prone to recall and/or reporting errors (41). For instance, pregnant women can unintentionally forget the exact number of minutes/hours they spend in PA or SB. Fur-the more, self-report questionnaires can underestimate SB (41) and overestimate PA (42). As such, the reported prevalence of compliance with SB and/or PA guidelines might have been under/overestimated. Thus, future research should consider using device-based approaches such as accelerometers to measure these movement behaviors in pregnant women. Another limitation is that the measurements were performed only at a single time-point during pregnancy while using a cross-sectional design, limiting our ability to infer longitudinal conclusions (43). Hence, further longitudinal studies that assess compliance with the 24-hour movement behavior guidelines in pregnant women throughout pregnancy are warranted to understand factors that influence patterns across pregnancy (44)."
- Comment: Methods: The recruitment process is poorly described, raising concerns about potential selection bias. There is no clear explanation of how the sample was representative of pregnant women in Saudi Arabia, nor how it addresses potential confounding factors such as socio-economic disparities.
Response:
Thank you for your comment. As the reviewer has suggested, we have added a new section to the materials and methods titled 'Recruitment and Participant Selection.' This section provides further clarification regarding the recruitment process, sample selection, and strategies implemented to mitigate selection bias.
"The recruitment and selection of the participants were performed using a thorough recruitment strategy as follows. First, all private and governmental hospitals and clinics offering obstetrics and gynecology services in six major cities in Saudi Arabia (Medina [West region], Riyadh [Central region], Al-Ahsa Governorate [East region], Jazan [South region], Jeddah [West region], and Taif [West region]) were reached. After obtaining the required permits, a research team member was stationed in waiting areas at obstetrics and gynecology clinics. Therefore, all women visiting these clinics were directly reached, with the intention of including pregnant women with diverse demographic and socio-economic statuses. All women who expressed interest in participation in the study were screened using the inclusion criteria. Thus, a wide spectrum of pregnant women who met the inclusion criteria were included in the study. This employed strategy has enhanced the diversity and repressiveness of pregnant women in Saudi Arabia"
- Comment: Methods: The study relies solely on self-reported questionnaires, which are known to be prone to recall and reporting biases. In the absence of objective measures (e.g., accelerometers), the validity of the findings is questionable.
Response:
Thank you for your comment. Please refer to the response to comment 2.
- Comment: Lack of Depth in Statistical Analysis: While the authors use prevalence ratios and regression models, there is limited explanation of how the models were adjusted for potential confounders. Moreover, with only 21 current smokers in the sample, it is highly questionable whether meaningful statistical conclusions can be drawn regarding smoking status. The paper would benefit from more advanced statistical techniques, such as multivariate analyses or sensitivity tests, to strengthen the reliability of the findings. The current analysis does not seem rigorous enough to support the conclusions drawn.
Response:
Thank you for your insightful comment regarding our choice of analysis. The primary objective of this study is to evaluate compliance with 24-hour movement behavior guidelines in pregnant women, identify non-compliant groups, and explore associated maternal factors. To achieve this, we employed a log-binomial regression model, a robust statistical technique well-suited for analyzing prevalence ratios. Furthermore, we acknowledge that the sample size of current smokers is relatively small, which might limit statistical power. However, we believe reporting the findings for this subgroup is valuable as it provides insights into the overall population. Moreover, our statistical strategy, as outlined in the analysis section, requires reporting and controlling for significant maternal characteristics, including current smoking status. Therefore, despite the smaller sample size, including data on current smokers, contributes to the overall understanding of our research question and its implications.
- Comment: Weak Discussion and Implications: The discussion is not well developed and does not sufficiently connect the evidence to implications for practice or policy. While the authors acknowledge that further interventions should be considered, they offer no strategies to integrate them into existing rehabilitation practices in Saudi Arabia. Moreover, they do not sufficiently relate their results to existing studies (especially other studies in the same facility or on similar populations). This diminishes the case for generalizability of their results.
Response:
Thank you for your comment. As detailed in the "Introduction" and "Discussion" sections, our study offers a novel contribution to the field by examining the prevalence of 24-hour movement behaviors in Saudi Arabian pregnant women. This is a relatively unexplored area, especially in the context of Saudi Arabia. To our knowledge, this study is the first to comprehensively evaluate adherence to all three components of 24-hour movement behaviors in pregnant women in Saudi Arabia. While the study offers valuable insights, it is important to acknowledge its limitations, particularly the cross-sectional design. The "Discussion" section addresses this limitation, along with other relevant factors, and compares our findings to existing literature, both locally and globally. This comparison provides a broader context for interpreting our results and identifies areas for future research.
- Comment: Conclusion Lacks Impact: The conclusion is limited to interpretation of results already presented in earlier sections, without new insights or concrete suggestions of where the results matter for public health practice. The study adds virtually nothing to the armamentarium of policy or clinical recommendations.
Response:
Thank you for your comment. As outlined in the "Clinical Significance" and "Conclusions" sections, the current study significantly contributes to the existing literature by shedding light on an overlooked aspect of 24-hour movement behaviors in pregnant women in Saudi Arabia. Our findings reveal, for the first time, that a majority of pregnant women in Saudi Arabia do not meet the recommended 24-hour movement behavior guidelines. Furthermore, we identified the first trimester as a particularly vulnerable period for adherence to these guidelines. The study also demonstrates that not all maternal characteristics are equally influential in determining compliance with 24-hour movement behaviors. Smoking status, region, and education emerged as the most significant factors influencing pregnant women's physical activity and sedentary behavior patterns. Please refer to the revised "Discussion" and "Conclusion" sections.
- Overall, the English used in the manuscript is acceptable but needs more clarity and flow, as well as professionalism. Although the authors are usually good at expressing their ideas, there are a few things that, in my opinion, make readability hard.
Response:
Thank you for your comment. Two coauthors, Abdullah Alansare and Bethnay Gibbs, made substantial contributions to the writing and review of the manuscript. Abdullah Alansare is proficient in English, having published many articles in the language. Bethnay Gibbs, a native English speaker, also provided valuable insights and revisions throughout the writing. Additionally, prior to submission, the manuscript underwent English proofreading by a third independent party (please see the attached supplement). Therefore, unless there is a specific issue with a particular sentence, we believe the manuscript is written in clear and correct English.
Round 2
Reviewer 1 Report
Comments and Suggestions for Authors
I would like to recommend the revised version of manuscript for publication. Authors used the comments given in my revision and completed the missing information in Participants and Procedure sections . In my opinion, now the research approach, methodology, interpretations and discussion are comprehensive